

# A review on cultivating effective learning: synthesizing educational theories and virtual reality for enhanced educational experiences

Fatma Mallek[1,*], Tehseen Mazhar[2,*], Syed Faisal Abbas Shah[2], Yazeed Yasin Ghadi[3] and Habib Hamam[1,4,5,6]

[1] Faculty of Engineering, University de Moncton, Moncton, Canada
[2] Department of Computer Science & Information Technology, Virtual University of Pakistan, Lahore, Pakistan
[3] Department of Computer Science and Software Engineering, Al Ain University, Abu Dhabi, United Arab Emirates
[4] Bridges for Academic Excellence, Tunisia, Tunisia
[5] School of Electrical Engineering, University of Johannesburg, Johannesburg, South Africa
[6] Hodmas University College, Mogadishu, Somalia
[*] These authors contributed equally to this work.

Corresponding authors
Tehseen Mazhar,
tehseenmazhar719@gmail.com
Syed Faisal Abbas Shah,
syedfaisalshah196@gmail.com

## ABSTRACT

Immersive technology, especially virtual reality (VR), transforms education. It offers immersive and interactive learning experiences. This study presents a systematic review focusing on VR's integration with educational theories in higher education. The review evaluates the literature on VR applications combined with pedagogical frameworks. It aims to identify effective strategies for enhancing educational experiences through VR. The process involved analyzing studies about VR and educational theories, focusing on methodologies, outcomes, and effectiveness. Findings show that VR improves learning outcomes when aligned with theories such as constructivism, experiential learning, and collaborative learning. These integrations offer personalized, immersive, and interactive learning experiences. The study highlights the importance of incorporating educational principles into VR application development. It suggests a promising direction for future research and implementation in education. This approach aims to maximize VR's pedagogical value, enhancing learning outcomes across educational settings.

# INTRODUCTION

## Context and motivation

In the world of education, changes are needed. Traditional ways often miss matching each student's unique way of learning. This can lead to problems like not having enough resources and tools to help every student in the way they need. *Yousafzai et al. (2021)* show how using deep learning can predict how well students will do, helping to tailor

education to their needs. It has been found that using social networks as learning tools can improve how students perform and enjoy their education (*Mejbri et al., 2022*). Meanwhile, *Saif et al. (2022)* explores how technology, especially ICT, is reshaping education, making it more accessible and effective. *Al-Shloul et al. (2024)* highlight how active learning and tools like ChatGPT can make learning more engaging and personalized. Combining these approaches could transform education, making it more flexible, interactive, and tailored to individual needs.

Education is pivotal in advancing individual growth and societal progress. However, conventional educational methods often struggle to address the diverse needs of students, the scarcity of resources, and the challenges in accurately assessing learning outcomes. The introduction of large language models (LLMs) into digital and smart education systems offers a promising avenue for improvement (*Joshi, Vinay & Bhaskar, 2021*; *Wensheng et al., 2023*). LLMs can process extensive datasets, identifying unique learning styles, preferences, and patterns among students, thus enabling the tailoring of educational experiences to individual needs (*Aggarwal, 2023*).

Artificial Intelligence (AI), including chatbots and virtual assistants like ChatGPT, plays a crucial role in providing students with immediate tutoring, feedback, and support, effectively supplementing traditional teaching methods and enhancing learning outside the classroom environment (*Javaid et al., 2023*; *Fitria, 2021*). These AI-driven tools not only facilitate instant responses to students' queries but also assist in the creation and organization of learning content, assessments, and resources, thereby enabling educators to devote more time to innovative and creative teaching methodologies (*Fitria, 2021*).

Additionally, the adoption of immersive technologies such as virtual reality (VR) transforms the educational landscape by offering students dynamic, interactive, and engaging learning experiences that are difficult to replicate in conventional classroom settings (*Dieck et al., 2023*; *Coyne et al., 2019*). VR fosters an environment where students can actively engage with the curriculum, providing hands-on experience in scenarios where real-world application may be impractical or hazardous (*De Luca & Fornatora, 2020*; *Killian et al., 2019*).

The effective incorporation of VR into educational strategies requires careful consideration of its alignment with educational goals and learning outcomes. The democratization of VR technology, through cost reduction, expands access to immersive educational experiences across various subjects, enhancing practical training and promoting innovation in teaching and learning practices (*Concannon, Esmail & Roduta Roberts, 2019*; *Pietroszek, 2019*). VR laboratories simulate real-world environments, allowing learners to experiment freely in a risk-free setting, thus addressing the limitations associated with physical laboratories and bridging the gap in distance learning (*Hernández-de Menéndez, Vallejo Guevara & Morales-Menendez, 2019*; *Childs et al., 2023*). By integrating LLMs and VR technology with educational theory, there lies a potential to significantly enhance the effectiveness, engagement, and personalization of the learning experience, heralding a new era in education. Figure 1 describes the different benefits of VR technology in the education sector.

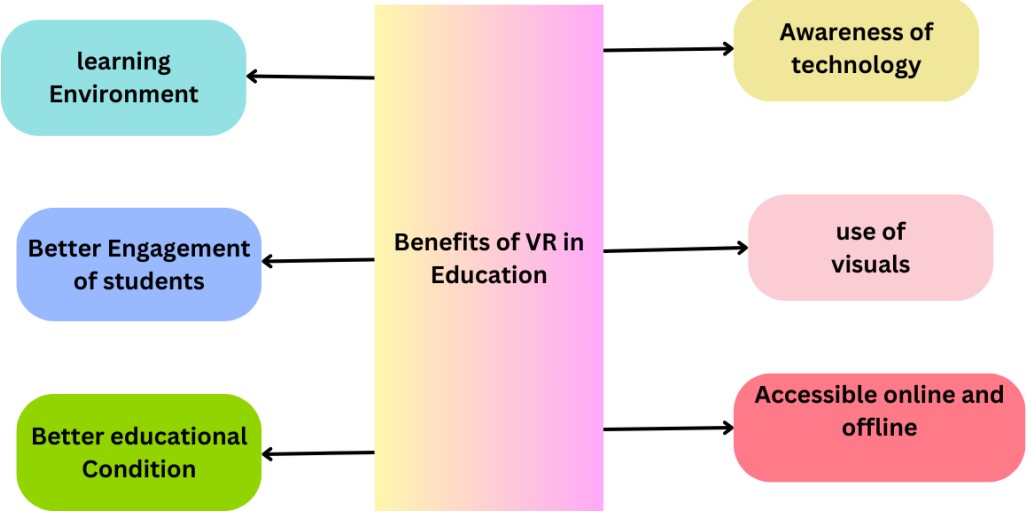

**Figure 1**  Benefits of VR in the education sector.

## Contribution of the study

Our study makes unique contributions to the scientific community, especially in the use of VR in higher education. We outline these contributions as follows:

- Comprehensive taxonomy: We introduce a new way to understand VR uses in higher education. We sort these uses into groups based on different ways of teaching and learning, like learning by doing, learning together, and building knowledge from experience. This sorting helps teachers and people who study education see the many kinds of VR tools available. It guides them in choosing the best VR methods to help students learn better.

- Systematic literature review: A big part of our work was looking carefully at what other researchers have found about combining VR with teaching theories in colleges and universities. We put all this information together to show what is new, what is missing, and what could be explored next in this area. This gives a complete guide for anyone interested in how VR and learning ideas come together.

- Comparative analysis: Our article compares old ways of teaching with new ways that use VR. We talk about the good things and the challenges of using VR in schools. This comparison helps show in real ways how VR can change and improve learning. It gives clear evidence of VR's impact on students' learning experiences.

- Framework for implementation: Lastly, we offer a step-by-step guide for bringing VR into educational settings. This guide is based on solid research and teaching theories. It shows educators how to use VR in ways that match what they want students to learn. It makes sure that VR tools used in teaching are based on good educational practices and help meet learning goals.

By focusing on these areas, our article adds new insights into the use of VR in education. It aims to help educators, researchers, and technology developers make learning more engaging and effective with VR.

## Structure of the article

This article unfolds in a structured manner, starting with an introduction that sets the stage for the subsequent exploration of VR in education. It begins by addressing the context and motivation behind using immersive technologies to enrich educational experiences, followed by a statement on the contributions this study aims to make to the existing body of knowledge.

- Section 'Research Methodology' delves into the research methodology, outlining the systematic approach adopted for the literature review, including search strategies, selection criteria, and data extraction methods. This section ensures transparency in how the studies were chosen and analyzed, providing a foundation for the review's findings.
- Section 'Literature Review' forms the core of the article, presenting a comprehensive examination of the integration of VR with educational theories in higher education. It is divided into sub-sections, each focusing on a different aspect of immersive technology applications in education, from enhancing practical skills and professional training to addressing the challenges and future directions in immersive technology education. This section synthesizes current knowledge, identifies gaps, and discusses the implications of VR in educational settings.
- Section 'Results' presents the results of the literature review, comparing traditional and VR learning approaches, and discussing the potential of VR laboratories in educational environments. This section highlights the advantages of immersive learning experiences and their impact on student engagement and learning outcomes.
- Section 'Discussion' provides an analysis of the evidence and findings from the literature review. It offers actionable insights for implementing VR in education, discusses the limitations of current VR applications, and suggests solutions to these limitations. This section aims to inform practitioners, researchers, and policymakers about the effective integration of VR into educational practices.
- The 'Conclusion and Future Recommendations' summarize the review's main points, reiterating VR's transformative potential in education and suggesting directions for future research. This section underscores the importance of continued exploration and innovation in the field of immersive technologies in education.

Throughout, the article integrates various figures and tables to illustrate concepts and summarize key information, enhancing the reader's understanding of the subject matter. The composition of this article provides a thorough and critical examination of the role of VR in enhancing educational experiences, offering valuable insights for educators, researchers, and technology developers. Table 1 illustrates the list of abbreviations.

**Table 1  List of abbreviations.**

| Abbreviations | Full form | Abbreviations | Full form |
|---|---|---|---|
| VR | Virtual reality | CL | Constructivist learning |
| AR | Augmented reality | CL | Collaborative learning |
| MR | Mixed reality | EL | Experiential learning |
| PBL | Problem-based learning | LMS | Learning management system |
| RQ | Research questions | ML | Machine learning |
| DL | Deep learning | LLMs | Large language models |

# RESEARCH METHODOLOGY

## Methodology and techniques

A literature review is commonly performed to identify significant gaps in the study or subject areas that have not been thoroughly researched, necessitating further investigation or analysis. This review is a valuable tool for identifying current research gaps and strategizing potential future research directions. The review requires a significant amount of time and effort as it involves evaluating the work of all academics who have contributed to various subjects up to this point. Several photo pieces related to the study were discovered during the preliminary search. By examining the availability of VR role in education material, one may come across a review article that has not been made public before. This can be attributed to the fact that VR is still a relatively new framework. This review has been completed with the help of the updated reference guide obtained from *Mealy (2018)*. Figure 2 illustrates the flow process of this article.

In developing our methodology for the systematic literature review, we utilized the comprehensive taxonomy introduced in 'Contribution of the study' as a foundational framework. This taxonomy, which classifies VR applications in higher education based on pedagogical theories such as constructivism, experiential learning, and collaborative learning, guided our search criteria and analysis. By aligning our review process with these categories, we were able to systematically identify and evaluate studies that demonstrate the diverse applications of VR, ensuring a thorough examination of how VR strategies align with different educational outcomes.

## Research objectives

This study aims to give academics and educators a complete review of contemporary research on educational theories and practices linked to the use of VR systems in education. Specifically, the focus of this overview will be on the use of VR systems in the education sector and laboratories. The purpose of this research is to provide a comprehensive analysis of the results.

## Research questions

1- What is the role of VR in the education sector and laboratories?
2- What are traditional and VR learning approaches in education?
3- What are traditional laboratories in educational environments and the potential of VR laboratories?

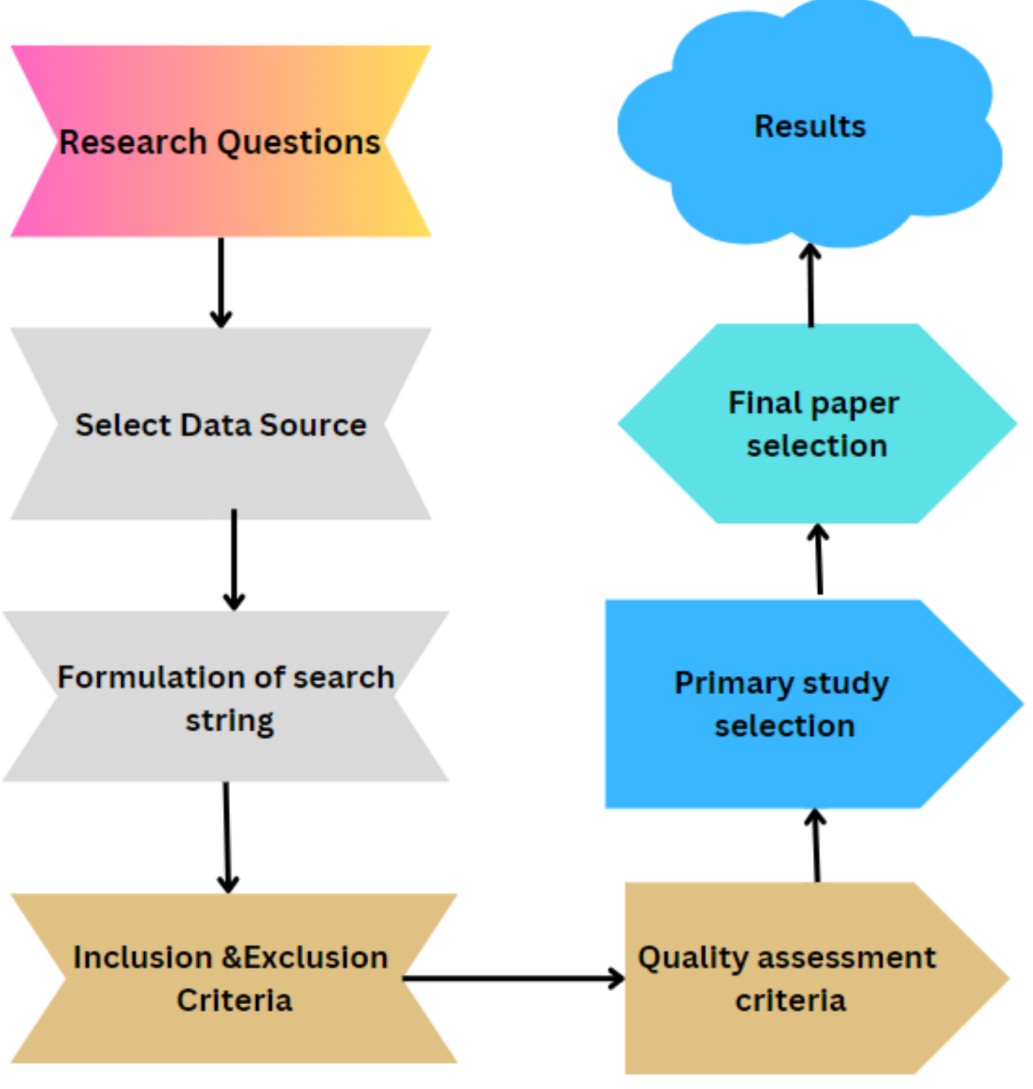

**Figure 2  Flow of work.**

4- What are the research gaps and why there is a need for the integration of learning theories and approaches in the development of VR laboratories?

5- What are the limitations of the implementation of VR in education?

## Search strategy

A keyword-based string that included VR methodologies in the education sector was used to search the publications in databases such as Springer, Scopus, A.C.M., Science Direct, Google Scholar, IEEE and Wiley. The following keywords were used to find the publications: immersive technology, VR, education sector, traditional laboratories, VR laboratories, educational theories and approaches. These studies, published in the journals mentioned above, were explored. The articles were examined after the original selection. The publications focusing on VR in education methodologies were then picked and

included in this study to master the fundamentals of VR. The publications were selected first by title relation to VR in education, then by complete filtering of the abstract, findings, and conclusions. The second screening step excluded many articles on VR applications in the medical field, education, business instruction, and defense training. All additional articles from the initial search were eliminated. We included only a subset of articles in the review because our goal was to establish a baseline of VR methodologies and research gaps to continue the study.

## Keyword searching

The final process of keywords, abstract, and title searching using different databases and articles inclusion-exclusion is shown in Fig. 3 and Table 2.

## Inclusion and exclusion

Our selection criteria focused on studies demonstrating VR's impact across various educational settings, particularly those offering insights into engagement, retention, personalization, safety, scalability, and collaboration in learning.

The term "inclusion criteria" refers to a set of rules that have been established and are applied in systematic literature reviews, or SLRs, to determine if research is eligible for inclusion in the review. The inclusion criteria listed below will be used in this analysis:

- The articles included in the study from 2016 to 2024 are a specific topic
- The articles in which the main focus of VR, AR, XR model evaluation, history, and training are included
- The review carried out in the education field will be centered on the use of AI, ChatGPT, and LLMs
- The selected study must focus on VR in educational learning environments
- The articles in which the main focus is VR, MR, XR in education, and some other fields are included in the study
- Only publications in reputable books, journals, or conferences should be accepted as sources for the articles that will be chosen.

The following categories of research have been recognized as potential candidates for removal from the investigation:
- Publications that were released in the years before 2016
- Studies where VR, AR, XR, or their application in education are not the primary focus
- Studies with poor numbers of results from empirical analysis
- Studies that do not evaluate the effectiveness of VR, AR, and XR implementation in learning environments

## Data extraction

Using Kitchenhands' rule as a guide, an Excel design was created at this point. The framework's primary objective is maintaining track of the data required to respond to the R.Q. This contains the outcomes of critical reviews as well as published works. The open article, article titles, keywords, abstracts, full texts, publication dates, fields, and other information are all included in the data gathered. The articles with "Yes" values in the three

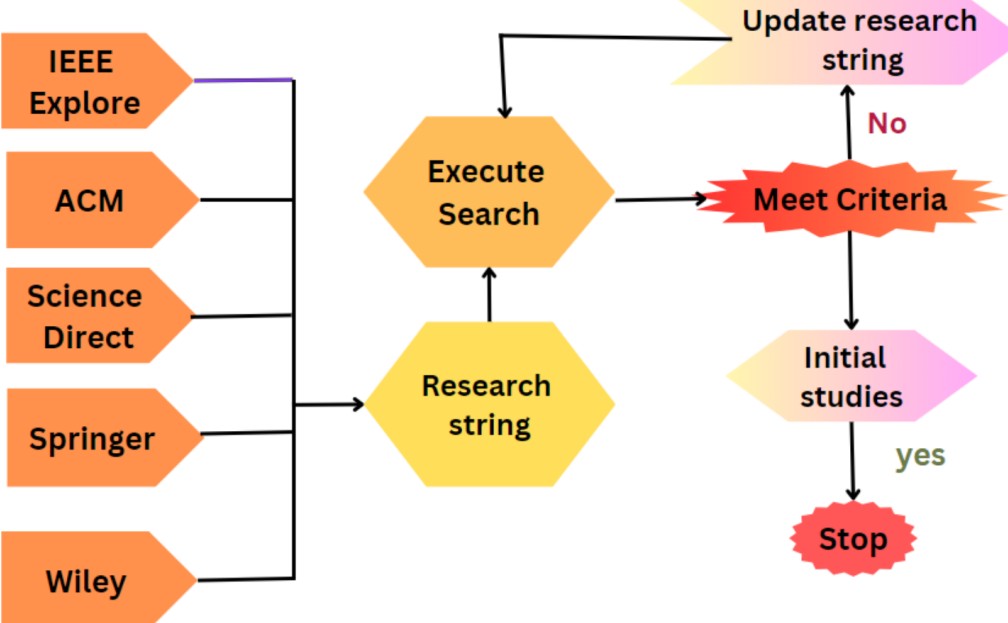

**Figure 3** Library and keyword selection process.

**Table 2** Primary and secondary keywords.

| Primary Keywords | Secondary keywords |
|---|---|
| Virtual reality, VR, Laboratories, Education, Educational Theories, Methods, MR, Challenges, Gaps, Teaching, Traditional, Integration, Students, Universities | Role, Types, Uses, VR, Education, Laboratories, Technology, Schools, Students. |

filter columns were transferred to a new sheet to obtain the data required to respond to the R.Q.s. An application is a text created by the same author that demonstrates practical applications of VR. "Implementation" refers to an article's explanation, description, comparison, or discussion of one or more VR implementation methods or algorithms. It explains how to use VR in education. The researcher studying VR in education will need to resolve a variety of issues.

## Article selection

The papers were selected from the different libraries based on title, abstract, and keywords. An initial search of the articles is shown in Fig. 4 and Table 3.

## Final article selection

The final articles were selected from the different libraries based on title, abstract, and keywords. The final search of articles is shown in Fig. 5 and Table 4. The prism diagram is shown in Fig. 6.

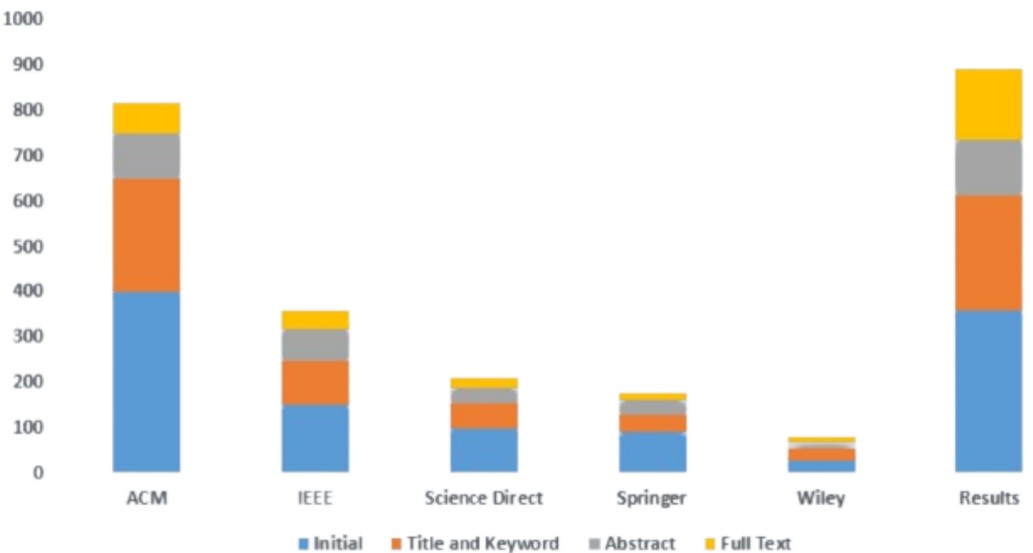

**Figure 4  Article selection.**

**Table 3  Article selection.**

| Library | Initial | Title and keyword | Abstract | Full text |
|---|---|---|---|---|
| ACM | 400 | 250 | 100 | 70 |
| IEEE | 150 | 100 | 70 | 40 |
| Science Direct | 100 | 55 | 35 | 20 |
| Springer | 90 | 40 | 32 | 15 |
| Wiley | 30 | 25 | 15 | 10 |
| Results | 770 | 470 | 252 | 155 |

# LITERATURE REVIEW

## Transformative learning through virtual and augmented realities

VR and augmented reality (AR) open new ways for students to learn. These technologies let students visit places far away, old historical sites, and areas that are hard to get to. Students do not need to leave their classroom to explore these places. They can see and interact with different settings and times through VR and AR (*Harknett, 0000*). This makes learning more interesting and real. For example, VR can show students what ancient Rome looked like or take them under the ocean to explore coral reefs. AR can bring a dinosaur into the classroom or show how a heart beats inside the human body. These experiences help students understand their lessons better because they can see and interact with what they are learning about (*Paraskevi et al., 2019*).

These technologies also help students learn things that are hard to do in real life because of safety or costs. VR and AR can simulate science experiments or let students practice fixing a car engine. This way, students can try many times until they learn how to do something without any risk (*Paulauskas et al., 2023*). Using VR and AR in education makes

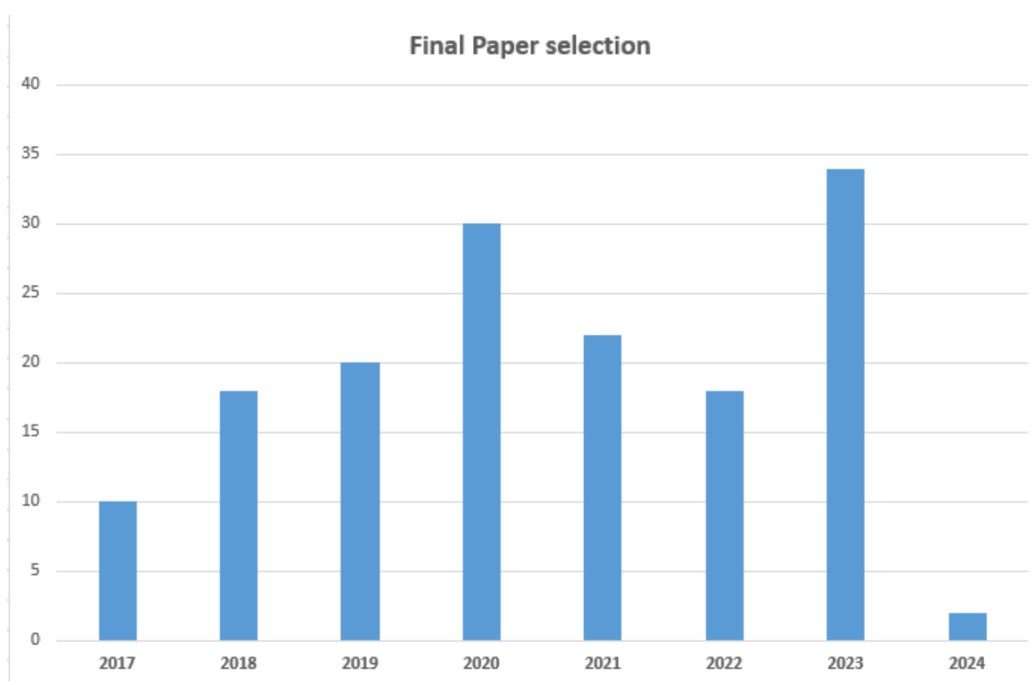

**Figure 5** Final article selection.

**Table 4** Final article selection.

| Year | No of papers | Year | No of papers |
|------|------|------|------|
| 2017 | 10 | 2021 | 22 |
| 2018 | 18 | 2022 | 18 |
| 2019 | 20 | 2023 | 34 |
| 2020 | 30 | 2024 | 2 |

learning exciting and helps students remember their lessons better. Teachers can use these tools to create amazing lessons that are fun and educational at the same time. This is a new way of learning that can help every student do better in school.

## Enhancing practical skills and professional training

In schools and training programs, learning by doing is very important. VR and AR help a lot with this. They let students practice skills in a world that feels real but is safe (*Poshmaal et al., 2021*). For example, someone learning to be a doctor can use VR to do a surgery without a real patient. This way, they can learn without any risk. VR and AR are also good for learning about cars, building things, and other jobs that need hands-on practice (*Garlinska et al., 2023*). Students can use VR to see inside a car engine or build a virtual bridge. They can try as many times as they need to learn how to do it right.

These technologies are not just for science and engineering. They help in many areas, like art, design, and language learning. Students can use AR to see art come to life or use VR

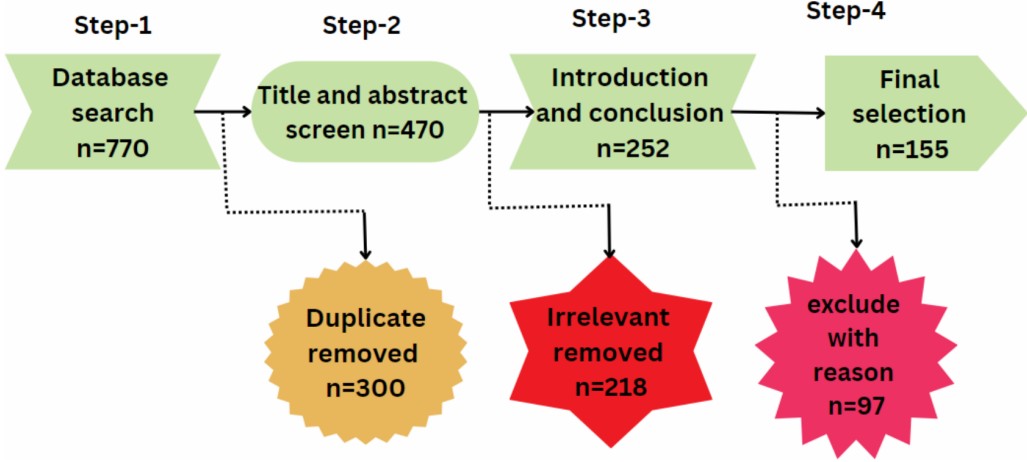

**Figure 6**  Prisma diagram.

to visit places around the world and practice speaking new languages (*Abid et al., 2022*). Training with VR and AR is very special because it can change as the student learns. If a student does well, the program can make the next lesson harder. If a student needs more help, the program can give more practice (*Mercan, Selçuk & Keskin, 2023*). This makes learning very personal and can help each student learn better and faster. Using VR and AR for learning skills is like having a personal teacher who can create any lesson needed. This is a big change in how people learn and can help students get ready for their future jobs.

## Fostering engagement and collaboration in digital learning environments

Learning together and talking to each other are big parts of school. VR and AR can make this even better, especially when students cannot be in the same place (*Xu et al., 2022*). These tools let students and teachers meet in a virtual world. It's like being in the same room, even if they are far away. For example, with VR, a whole class can visit a virtual museum together. They can see and talk about the art or science exhibits as if they were walking through the museum in real life. This helps students learn from each other and build teamwork skills (*Makhataeva & Varol, 2020*).

AR can help students work on projects together in a new way. They can see the same 3D model of a building or a molecule on their desks, even if they are not in the same place. They can change it, fix it, and make it better together (*Li, Fahmy & Sienz, 2019*). This kind of teamwork on projects is very important for learning and getting ready for the future.

Teachers can use VR and AR to make lessons that are fun and bring students together. They can create games where students have to work as a team to solve a puzzle or complete a mission. This makes learning exciting and teaches students how to work with others (*Wonsick & Padir, 2020*). Also, VR and AR can help students who feel shy or are afraid to speak up in class. In a virtual world, they might feel more comfortable sharing their ideas. This is good because every student has something important to say (*Prati et al., 2021*).

Using VR and AR for working together and talking to each other in school is a great way to make learning fun and help students get ready for the world outside of school.

### The role of mixed reality in education and training

Mixed reality (MR) mixes the real world with virtual worlds to create a new place where students can learn by seeing and doing. MR lets students and teachers see and use things that aren't there but feel like they are. This is great for learning because it makes hard ideas easier to understand (*Sievers et al., 2020*). For example, in science class, students can see how atoms come together to make things, or in history class, they can walk through ancient cities. They can look around, move things, and see how everything works from all sides. This helps students learn better because they can see what they are learning in a way that feels real (*Frank, Moorhead & Kapila, 2017*).

MR is also very useful for people learning to fix cars, fly planes, or do surgery. They can practice these skills in a place that feels real but is safe. If they make a mistake, they can try again without any real-world problems. This kind of practice makes learning faster and helps students feel more confident (*Zhang et al., 2020b*).

Teachers can use MR to make lessons that reach every student. Some students learn best by reading, some by listening, and some by seeing or doing. MR can help with all these ways of learning. It can show videos, play sounds, and let students interact with lessons in a way that works best for them (*T et al., 2017*).

MR can also help students work together, even if they're not in the same place. They can see the same things and work on the same problems in the MR world. This helps students learn how to work in teams and solve problems together, which are important skills for the future (*Nakamura et al., 2020*). In the end, MR can change how we learn by making it more active, fun, and real. It helps students understand complex ideas, practice skills safely, and work together in new ways. This prepares them for a world where technology is a big part of everything we do (*Whitney et al., 2018*).

### Addressing challenges and future directions in immersive technology education

Using VR, AR and MR in schools is very exciting. However, there are some challenges too. One big challenge is making sure all students can use these technologies. Not all schools have the same amount of money or equipment (*DelPreto et al., 2020*). Teachers need to learn how to use these new tools too. They need time and training to make the best lessons with VR, AR, and MR (*Shi et al., 2023*).

Even with these challenges, the future looks bright. More and more, teachers and students are using these technologies to learn in new ways. As more people see how helpful VR, AR, and MR can be, more schools will want to use them. This means companies will make more and better equipment that is easier to use and costs less money (*Pendergast et al., 2022*). In the future, lessons with VR, AR, and MR could become even more like real life. This would make learning even more exciting and helpful. For example, students could practice being doctors with VR that feels just like being in a hospital. Or, they could use AR to see how buildings are made, layer by layer, right on their desks (*Zong & Krishnamachari, 2022*).

Students could also meet in virtual classrooms from anywhere in the world. They could learn about other cultures by "visiting" other countries with VR. They could talk to students from those countries as if they were there in person. This would help students learn how to work with people from all over the world (*Zhang et al., 2020a*). Even with the challenges, using VR, AR, and MR in education has a lot of promise. These technologies can make learning more fun and help students get ready for a world where technology is everywhere. Teachers, students, and technology makers need to work together to solve the challenges. Then, the future of learning with these amazing tools can become a reality for all students (*Kasneci et al., 2023*).

## Emerging trends and research in immersive educational technologies

The world of learning is always growing, especially with new tech like VR, AR and MR. These tools are not just about seeing and doing things in cool, new ways. They are about making learning better for everyone (*Hadi et al., 2023*). Researchers are finding new ways to use VR, AR, and MR in class. They are looking at how these tools can help students understand tough subjects by letting them "jump into" what they are learning (*Lo, 2023*). For example, students might use VR to walk around ancient Egypt or use AR to see how a plant grows from a seed, all from their classroom.

One big trend is combining VR with AI. This mix can make learning very personal. AI can change what a student sees and does in VR based on what they need to learn next (*Dwivedi et al., 2023*). It's like having a teacher just for you, who knows exactly what you need to learn and the best way for you to learn it. Another exciting area is using VR and AR to help students work together, even if they are far apart. This can make learning with friends from around the world possible. It's a great way to learn about other cultures and ways of life (*Lund & Wang, 2023*).

There is also a big focus on making VR and AR easy for everyone to use. This means making them less expensive and easier to get. It also means teaching teachers how to use these technologies so they can bring them into their classrooms (*Rossoni et al., 2024*). Studies are showing that VR, AR, and MR can help students learn better. They can make complicated ideas easier to understand and remember (*Dai et al., 2020*). They can also make students more excited about learning and more willing to try new things (*Biswas, 2023*).

In the future, learning with VR, AR, and MR could become even more amazing. We might see students doing things like practicing surgeries, exploring the ocean, or designing their cities with these tools (*Abramski et al., 2023*). The possibilities are endless. But to get there, everyone needs to work together—teachers, students, and the people who make these technologies. By sharing what they learn and helping each other, they can make learning with VR, AR, and MR the best it can be for everyone (*Sobieszek & Price, 2022*).

In the evolving landscape of education, immersive technologies such as VR, AR, and MR are not just tools for engagement but are becoming fundamental in shaping the educational paradigms of the future. Research and development in these areas are uncovering new potentials for learning environments, teaching methodologies, and student interactions (*Cai, Van Joolingen & Walker, 2019*). Recent studies have explored how VR can simulate

complex biological processes, allowing students to visualize and interact with cellular mechanisms in a way that textbooks cannot convey (*Mazhar et al., 2023*). This immersive approach has been shown to significantly improve understanding and retention of complex subjects.

In the domain of history and archaeology, AR applications are bringing ancient civilizations to life, enabling students to explore historical sites and artifacts in unprecedented detail. This has the potential to foster a deeper appreciation and understanding of historical contexts and cultures (*Koolivand et al., 2024*). Advancements in MR technology are bridging the gap between digital and physical learning environments. For example, MR has been utilized in engineering education to overlay digital information onto physical objects, such as machinery parts, facilitating a hands-on learning experience that integrates theoretical knowledge with practical application (*Fitria, 2023*).

Collaborative learning has also been enhanced through immersive technologies, with platforms allowing students to work together in virtual spaces from different geographical locations. These virtual collaborative environments have been particularly effective in disciplines requiring team-based projects, such as architecture and design (*Sun et al., 2023*). The integration of AI with immersive technologies is paving the way for personalized learning experiences. AI algorithms can adapt the learning environment in real-time, based on the student's performance and engagement, offering tailored support and challenges to optimize learning outcomes (*Smutny, 2023*).

Efforts are also being made to address accessibility and inclusivity within immersive learning environments. Technologies are being developed to ensure that students with disabilities can fully participate in VR and AR experiences, making education more equitable (*Mokmin et al., 2023*). The potential of immersive technologies in professional and vocational training is being realized through simulations that provide learners with realistic, risk-free environments to practice and hone their skills. This is particularly valuable in fields such as medicine, aviation, and automotive industries, where practical experience is crucial (*Aguayo & Eames, 2023*).

However, the implementation of these technologies in education faces challenges, including high costs, technological literacy, and the need for comprehensive curricular integration. Ongoing research is focused on overcoming these barriers, ensuring that the benefits of immersive learning can be accessed by a broader audience (*Hmoud et al., 2023*). As we look to the future, the trajectory of immersive educational technologies is set toward more interactive, collaborative, and personalized learning experiences. The continued collaboration between educators, technologists, and researchers is crucial in harnessing the full potential of these technologies to revolutionize education (*Marougkas et al., 2023*).

## Case studies and longitudinal insights in immersive technology education

The integration of immersive technologies into education has been the subject of various case studies and longitudinal research, providing valuable insights into their effectiveness, challenges, and future directions. These studies offer a closer look at real-world applications and their impact on learning outcomes and pedagogical approaches. For instance, a

multi-year study involving VR in biology education demonstrated not only an increase in student engagement and motivation but also a significant improvement in test scores, underscoring the potential of immersive technologies to enhance academic performance (*Aebersold, Rasmussen & Mulrenin, 2020*). Similarly, AR has been utilized in language learning programs, where it contributed to higher levels of student interaction and confidence in using a new language in practical settings (*Luan & Tsai, 2021*).

In the field of environmental science, MR applications have enabled students to virtually explore ecosystems and biodiversity, fostering a deeper understanding of ecological principles and environmental stewardship (*Khanzode & Sarode, 2020*). These experiences have been shown to promote active learning and critical thinking skills, essential competencies in today's rapidly changing world. A notable case study in the domain of history and cultural education involved the use of VR to recreate historical sites and events, allowing students to 'experience' history rather than merely reading about it. This immersive approach has led to a more profound connection with historical content and a better appreciation of cultural heritage (*Chassignol et al., 2018*).

Technical and vocational training has also benefited from immersive technologies, with VR simulations providing a safe and controlled environment for practicing complex procedures in healthcare, automotive repair, and industrial maintenance. These simulations have been effective in reducing learning curves and improving procedural accuracy (*Abideen et al., 2023*).

In the arts, AR and VR have been harnessed to create interactive exhibitions and performances, offering new ways for students to engage with art and design concepts. These technologies have opened up new avenues for creative expression and interpretation, highlighting the interdisciplinary potential of immersive technologies (*Huang et al., 2021*). Despite these successes, challenges such as technological accessibility, cost, and the need for specialized training for educators remain. Future research is directed towards addressing these barriers and exploring scalable models for integrating immersive technologies into mainstream education (*Gray & Perkins, 2019*).

The ongoing evolution of VR, AR, and MR technologies promises to continue shaping the educational landscape. As these technologies become more sophisticated and accessible, their potential to transform learning environments and educational methodologies will only increase. Collaborative efforts among educators, technologists, and policymakers are essential to fully realize this potential and ensure that immersive technologies contribute to equitable and effective education for all learners (*Iqbal et al., 2017*).

Furthermore, a broad spectrum of research has focused on enhancing collaborative skills and empathy among students through VR experiences, with studies showing that virtual environments can significantly impact students' ability to work in teams and understand diverse perspectives. These findings are elaborated across multiple disciplines and settings (*Zhang et al., 2018*; *Fu et al., 2018*; *Rastrollo-Guerrero, Gómez-Pulido & Durán-Domínguez, 2020*; *Suri et al., 2023*).

In the domain of specialized education, such as medical and healthcare training, AR and VR technologies have been pivotal in creating simulations for surgical procedures and patient interactions, providing a hands-on learning experience without the real-world

risks. This area of application is thoroughly investigated in several case studies and research articles (*Pottle, 2019*; *Huber et al., 2017*; *Sureephong et al., 2023*). The application of MR in engineering and architecture education has also been a subject of considerable interest. By blending digital elements with real-world environments, MR facilitates a deeper understanding of complex concepts and designs, as discussed in numerous studies (*Sala, 2021*; *Stepan et al., 2017*).

Emerging technologies in immersive learning environments have opened new avenues for experiential learning, where students can engage in simulated ecosystems, global markets, and virtual labs. These innovative approaches to education are explored in depth in recent literature (*Halbig et al., 2022*; *Cai, van Joolingen & Veermans, 2021*; *Ding & Li, 2022*). Additionally, the effectiveness of immersive technologies in increasing student motivation and engagement has been a key focus of educational research. Studies have documented significant improvements in learning outcomes and student satisfaction when immersive tools are integrated into the curriculum (*Shaytura et al., 2021*; *Farah, Ramadan & Harb, 2019*; *DuBose, 2020*).

The integration of immersive technologies in distance learning and remote education has also been critically examined. With the increasing need for accessible and flexible learning solutions, VR, AR, and MR have been identified as valuable tools in bridging geographical gaps and enhancing the online learning experience, as highlighted in several studies (*Chiang, 2021*; *Suen, Chiu & Tang, 2020*; *Hannah, Huber & Matei, 2019*).

Lastly, the future of immersive technologies in education looks toward the development of more inclusive and adaptive learning environments. Research continues to explore how these technologies can be tailored to meet diverse learning needs and styles, ensuring equitable access to quality education for all students (*Çankaya, 2019*; *Mealy, 2018*; *Wallgrün et al., 2019*).

# RESULTS

Our findings substantiate the utility of our taxonomy, revealing a broad spectrum of VR applications across higher education that align with the identified pedagogical theories. The taxonomy proved instrumental in categorizing the varied uses of VR, from immersive simulations enhancing experiential learning to collaborative VR projects fostering teamwork and communication skills among students. This classification not only highlights the multifaceted nature of VR in education but also illuminates specific areas where VR strategies have been particularly effective in enhancing learning outcomes.

## Traditional and VR learning approaches in education

This part describes how VR can be applied to education. The traditional learning approach refers to the conventional methods of teaching and learning that have been widely used in educational settings for many years (*Keele, 2007*). On the other hand, the VR learning approach incorporates VR technology to enhance the learning experience. Table 5 illustrates the disadvantages of traditional learning and VR learning methods.

**Table 5  Disadvantages of traditional and VR learning methods.**

| Disadvantages | Traditional learning methods | VR learning methods |
|---|---|---|
| Cost | Limited cost for materials and facilities. | High initial investment in hardware, software, and content. |
| Technical issues | Relatively stable and mature technology. | Prone to technical glitches, and hardware malfunctions. |
| Learning curve | Familiar and established teaching methods. | The steep learning curve for users and educators. |
| Accessibility | Limited accessibility for remote or disabled learners. | Potential barriers for those with disabilities. |
| Interactivity | Limited interactivity and engagement. | Enhanced interactivity, but may lack human interaction. |
| Personalization | Difficulty in personalizing learning experiences. | Potential for personalized learning experiences. |
| Resources | Reliance on physical resources like textbooks. | Dependence on digital resources and equipment. |
| Isolation | Opportunities for face-to-face interaction. | Potential for isolation due to immersive environments. |
| Health concerns | Minimal health risks associated. | Health risks such as eye strain or motion sickness. |
| Adaptability and scalability | Can be adapted to different environments easily. | May require significant adjustments for different subjects. |
| Real-world application | Immediate application in real-world settings. | Limited real-world applications outside of simulations. |

## Traditional learning methods

In the traditional approach, the teacher takes on a central role as the primary source of knowledge and instruction. The teacher delivers lectures, presents information, and guides students through learning (*Eze, Chinedu-Eze & Bello, 2018*). Traditional teaching primarily occurs in physical classrooms, where students gather to receive instruction from the teacher. Face-to-face interaction is a significant aspect of the traditional approach. The curriculum is typically predetermined and structured, often following a set syllabus. The teacher follows a prescribed sequence of topics and materials. Traditional teaching methods may have limited integration of technology as described in Fig. 7. While tools like whiteboards, projectors, and audiovisual aids are commonly used, extensive use of digital technology and online resources may be limited (*Saini & Goel, 2019*). Traditional methods often promote passive learning, where students play a relatively passive role, listening to lectures, reading textbooks, and taking notes (*Saini & Goel, 2019*). This passive approach may limit active engagement, critical thinking, and a deeper understanding of the subject matter. Traditional methods rely heavily on physical resources such as textbooks and printed materials. This limitation can restrict access to up-to-date information, diverse learning resources, and real-time data readily available through digital technologies (*Tao et al., 2019*).

*Fatani (2020)* shows that many universities are carefully planning and designing online courses to ensure that they provide meaningful learning experiences and support student success. Learning management systems (LMS), video conferencing platforms, collaborative document editing, and other software enable seamless content delivery, communication, and resource sharing. Blended learning is an approach that offers the benefits of both face-to-face interactions and the flexibility and interactivity of online learning (*Daskan & Yildiz, 2020*). By combining these approaches, educators can provide a more personalized and engaging learning experience, accommodating diverse learning styles and needs. The shift from traditional to online learning can bring several benefits to students. Students can access course materials, participate in discussions, and complete assignments at their convenience, allowing them to tailor their learning to fit their schedules and obligations.

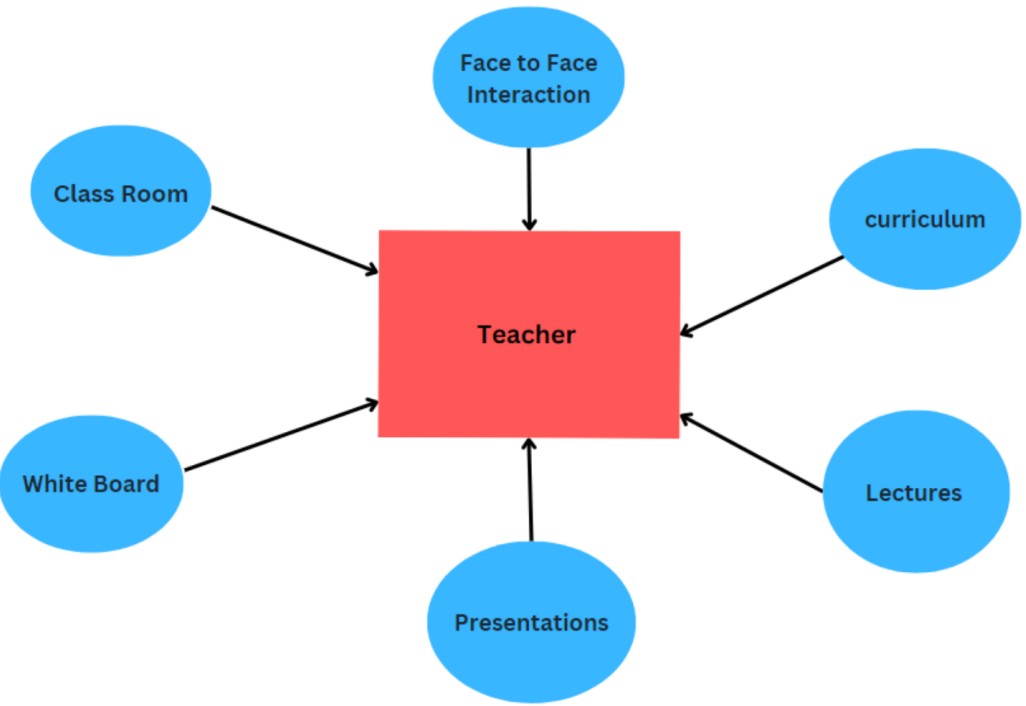

**Figure 7  Traditional learning methods.**

Online learning allows students to connect with peers and instructors from diverse backgrounds and geographical locations. This global reach fosters exchanging ideas, different cultural perspectives, and project collaboration (*Hilliker, 2020*).

## VR learning methods

VR learning methods offer numerous benefits, including increased engagement, enhanced retention of information, and the ability to practice skills in realistic environments. VR learning methods in education create immersive and interactive learning experiences (*Pellas, Dengel & Christopoulos, 2020*). Students can simulate complex scenarios and can get training in various fields. VR can replicate laboratory settings, allowing students to conduct experiments and practice scientific procedures virtually. According to the study *Al-Samarraie & Saeed (2018)* VR facilitates collaborative learning experiences by enabling students to interact with each other in a shared virtual space. Students can work together on projects, solve problems, and engage in group discussions, regardless of their physical location. *Ogbuanya & Onele (2018)* developed a VR application, which proved to be quite effective in elevating their student's level of comprehension regarding difficult fluid mechanics issues as mentioned in Fig. 8.

According to *Loucif et al. (2021)* several studies have been conducted in the UK exploring the Use of VR in teaching and learning contexts. *Fertleman et al. (2018)* describes that the University of College London (UCL) study investigated the impact of VR on learning outcomes and engagement in science education. It found that students who used VR simulations had significantly higher knowledge retention and engagement than traditional

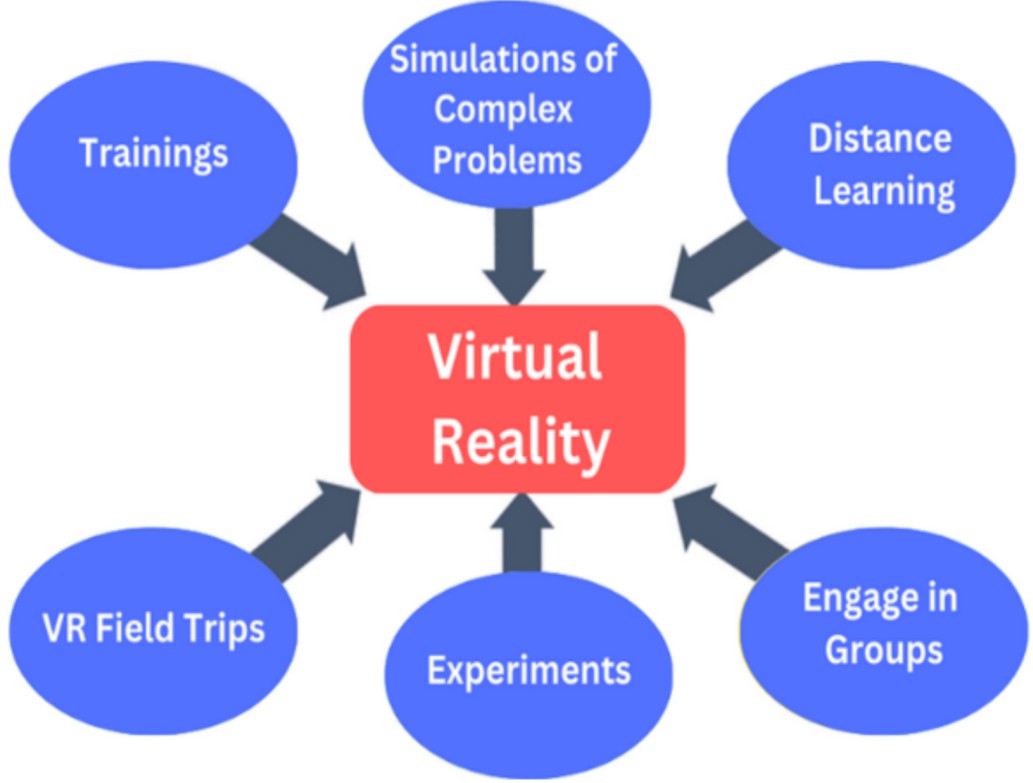

**Figure 8** **VR learning methods.**

methods. *Parong & Mayer (2018)* examined the effectiveness of VR in teaching physics concepts to secondary school students. It found that VR simulations improved students' conceptual understanding and engagement, particularly in complex and abstract topics. 3D VR technologies can enhance student outcomes, improve conceptual understanding, and provide transformative learning experiences. 3D VR allows the visualization of abstract and complex concepts that may be challenging to grasp through traditional methods (*Slavova & Mu, 2018*). By representing concepts in three-dimensional space, students can observe and manipulate objects, models, and data, facilitating a deeper understanding of the subject matter (*Fogarty, McCormick & El-Tawil, 2018*). When compared to seeing the experiment unfold on a projector in a classroom, the results of the experiment (*Schöne, Wessels & Gruber, 2019*) reveal that using VR results in an autobiographical recollection that is comparable to experimenting with real life. This brings up the point about how important laboratories and practical activity are in education.

## Traditional laboratories in educational environments and the potential of VR laboratories

Laboratories play a crucial role in education, providing students with unique opportunities to enhance their learning experiences and develop a deeper understanding of various subjects (*Viegas et al., 2018*). Laboratories in universities often use traditional methods

to conduct experiments, which have been used for many years. Manual setups, data collection, and calculations can be time-consuming, limiting the number of experiments students can perform within a given period. VR learning in education laboratories can revolutionize how students experience and engage with scientific concepts. In VR laboratories, students can conduct experiments without the risks associated with real-world laboratories (*Muradova, 2020*). This part discusses the differences and benefits of both traditional and VR laboratories. Table 6 highlights the drawbacks of VR labs in educational environments, problem-based learning (PBL) methods, and collaborative learning.

## The role of traditional laboratories in educational environments

The role of laboratories in education is multifaceted and crucial in providing students with practical learning experiences. Laboratories are vital in enhancing learning and fostering a deeper understanding of various subjects. In lab work, we combine problem-based with project-based learning, students gain a deeper understanding of the subject matter, develop valuable skills, and experience the satisfaction of applying their knowledge to real-world challenges. It promotes an active and engaged learning environment, preparing students for future academic and professional endeavors (*Virtue & Hinnant-Crawford, 2019*). Laboratories provide opportunities for students to gather empirical evidence. Students can validate or refute theories they have learned in the classroom by conducting experiments and collecting data. Laboratories often involve the Use of hazardous chemicals, equipment, and materials. Without proper safety measures, there is a risk of accidents, and injuries, to dangerous substances (*Salazar-Escoboza et al., 2020*). In physical laboratories, educational institutions may face budget constraints, limiting the availability of necessary resources for conducting experiments effectively. Scaling up the laboratory experience for larger classes may pose logistical challenges and reduce the quality of hands-on learning opportunities (*DeBoer et al., 2019*). According to Henderson and the United Nations Working Group on Habitat III, the education students get at universities is impeded by safety variables, lack of adequate facilities and machinery, and limited time and space accessibility (*dela Cruz & Mendoza, 2018*). *AlAwadhi et al. (2017)* adds that harmful errors in chemical interactions and electrical experiments can result in disastrous consequences and that pricey materials are a barrier to particular universities.

## The role of VR laboratories in educational environments

To overcome obstacles in traditional laboratories, VR has the potential to overcome various obstacles faced by traditional laboratories, bringing about significant improvements in research, training, and collaboration. *Zinchenko et al. (2020)* suggests that using VR to create virtual laboratories enhances students' learning experiences and increases their knowledge levels. Innovative teaching methodologies development is needed to enhance learning experiences and improve student engagement. Active learning often requires access to modern technology, interactive tools, and collaborative spaces. However, not all schools and universities in the UK may have the necessary infrastructure to support active learning effectively (*Børte, Nesje & Lillejord, 2023*). VR offers several benefits to

**Table 6  Drawbacks of VR labs in educational environments.**

| Drawbacks | VR laboratories in educational environments | Problem-based learning approach | Collaborative learning approach |
|---|---|---|---|
| Cost | High initial investment in VR hardware and software. | May require additional resources for problem development. | May need technology for collaboration tools. |
| Technical issues | Prone to technical glitches and compatibility problems. | Complexity in designing effective problems. | Technical issues with collaborative software or platforms. |
| Learning curve | Steep learning curve for educators and students. | Requires training for both educators and students. | Individuals may struggle with teamwork or coordination. |
| Accessibility | Accessibility challenges for students with disabilities. | May not cater to all learning styles or needs. | Potential barriers for students with limited access to tech. |
| Interactivity | Enhanced interactivity but may lack real-world context. | Dependent on problem quality for engagement. | Requires active participation for effective learning. |
| Personalization | Potential for personalized learning experiences. | Difficult to personalize learning for each student. | Balancing individual and group needs can be challenging. |
| Realism | Limited realism compared to real-world labs. | May lack real-world authenticity in problem scenarios. | May not fully replicate real-world collaboration dynamics. |
| Assessment | Challenges in assessing student performance accurately. | Assessment may be subjective and time-consuming. | Difficulty in assessing individual contributions fairly. |
| Time constraints | Limited time for hands-on experience in VR labs. | PBL sessions may require extensive time commitment. | Time-consuming coordination and scheduling for group work. |
| Maintenance | Requires regular maintenance and updates for VR equipment. | Continuous development and refinement of problem sets. | Need for ongoing facilitation and monitoring. |

distance learning students. VR enables active learning through interactive simulations and activities. VR platforms can facilitate social interactions and collaboration among distance learning students. They can engage in group projects and discussions. Creating a virtual laboratory is rapidly becoming one of the most important educational technologies in developing academics and technology. It is possible that the virtual laboratory will replace the traditional laboratory and that similar tests can be performed in two- or even three-dimensional simulated virtual settings. Using the virtual laboratory discovers a cost-effective option for schools and universities and a great instrument for distance learning education (*Eljack, Alfayez & Suleman, 2020*). *Makransky et al. (2021)* come to a comparable conclusion that learning through videos and learning in VR are both equally effective, which is that there is no distinction among the these. This lends credibility to the idea that VR can provide students who participate in distance learning with an education and learning experience on a level with those who attend classes on campus full-time. According to *Radhamani et al. (2021)*, Recently, to determine the extent to which students in the mechanical engineering discipline are learning from virtual labs, a training course was run for instructors in the scientific engineering discipline. These teachers subsequently instructed students in fluid mechanics in virtual laboratories. This was done to measure the learning level of virtual laboratories in the mechanical engineering discipline. The vast majority of educators expressed support for the role that virtual laboratories can serve in improving their teaching abilities and assisting students in completing their laboratory practices without adversely affecting the quality of students'

education. According to research conducted by *Diwakar et al. (2019)*, CL can be facilitated by using virtual laboratories that include pre- and post-lab sessions.

An organic chemistry VR laboratory was developed, as stated by *Ramírez & Bueno (2020)*, and students' short-term and long-term memory were compared to those of traditional laboratories. The findings demonstrate that there is no significant difference in the learning outcomes and the student's thoughts in both areas, which suggests that students participating in distance learning can benefit from being able to do the laboratory experiments while getting the same educational opportunity as full-time on-campus students. *Xie, Ryder & Chen (2019)* shows that mobile-based VR applications are rapidly being implemented in various educational settings, and in-depth assessments of how students engage with these tools are essential for the future design and study of VR. Knowledge of the boundaries and consequences of using these technologies within higher education has considerable relevance for instructors, professionals, and instructional designers for the classroom. This technology is new and has the potential to be very effective. The benefits of VR laboratories extend beyond the university or institution to the student's learning expertise, thinking, and pedagogical. Evidence from several study articles that produced and deployed VR-enabled laboratories demonstrates the technology's potential and excellence in education. *Pham et al. (2018)* created and verified a VR program that simulates a construction site, allowing students to immerse themselves in that environment remotely in a safe classroom setting to enhance their practical and secure experience. The study also shows that VR may be a powerful pedagogical tool for improving student learning. *Gargrish, Mantri & Kaur (2020)* created an AR application to help pupils visualize and grasp complicated 3D geometry. Although the program is built on AR, it is a visualization tool comparable to VR that emphasizes the potential advantages and uses of these tools. The author of *Yang et al. (2017)* Created a VR application for the ancient Great Wall of China construction methods, saying that VR has yet to catch up in beneficial engineering education areas. The capacity to see such old structures is crucial for teaching, and it demonstrates the promise of VR technology in education. Table 7 illustrates the advantages and disadvantages of VR in education.

## Research gaps and integration of learning theories and approaches in the development of VR laboratories

As mentioned in the above articles, VR is increasingly being adopted in educational settings for both students and professionals. If learning theories are not considered during the design of a VR application, several drawbacks and limitations can arise, impacting the effectiveness and overall learning experience. The research of *Drakatos et al. (2023)* points out inadequate research that determines suitable teaching or learning theories for training and education. They also point out that more initiatives and research must take place to evaluate VR technology with emerging methods of instruction and learning. This article looks at implementing learning theories and approaches to increase confidence in VR technology's ability to maximize students' engagement, learning experience, and theoretical thinking. By integrating these educational learning theories and approaches into VR laboratories, educators can create immersive and effective learning environments

**Table 7  The advantages and disadvantages of VR in education.**

| Advantages | Disadvantages |
| --- | --- |
| It enables the creation of complex test scenarios, experiments, and experiments that are difficult to implement in a real-world setting. | Costs are associated with creating an appropriate educational station using VR technology based on professional hardware and software. |
| Enables one to gain confidence in implementing technical procedures and activities. | Requires a lot of work to create a virtual environment with many test scenarios and details. |
| Allows for multiple repetitions of experiences, experiments, or situations. Limited scope or lack of ready-made teaching scenarios | Limited scope or lack of ready-made teaching scenarios |
| Saves money and time associated with setting up actual test stations. | No real consequences for mistakes and errors made. |
| Allows to perform exercises at any place and at any time. | The ability to make users addicted to the virtual world. |
| Ensures scalability of educational activities. | Limits interpersonal contacts and experiences. |
| Reduces consumption of real resources. | High probability of acquiring routine in the actions taken. |
| Ensures the safety of operations. | Potential for health problems for users. |
| Has the ability to adapt and apply to various fields and areas of education. | The possibility of ignoring basic laws of physics |
| Increases the ability to communicate and collaborate with people in remote locations. | It reproduces better or worse the given reality but is not able to replace it entirely |

that cater to diverse learning styles, promote active engagement, and facilitate meaningful learning experiences for students.

## Problem-based learning approach

The recent progress in technology capabilities has given educators an abundance of chances to use creative teaching strategies to maximize student learning. Problem-based learning (PBL), is still the primary instruction method. Students' knowledge, psychomotor abilities, interpersonal skills, self-confidence, and motivation can all benefit from the beneficial effects of VR combined with PBL (*Indriani, Isnarto & Mariani, 2019*). Integrating PBL principles into the design and development of VR experiences empowers learners to develop practical skills and knowledge directly applicable to real-world situations and foster a deeper understanding (*Smith et al., 2022*). In *Chang et al. (2022)*, the authors designed a VR project practical course that leads students in cooperative learning with their peers, inspires students to explore and solve challenges actively, and provides real-time intelligent feedback to participants. Most students' scores on the post–test measuring the learning efficiency of the PBL VR course in situated learning are much higher than those on the corresponding pre-test.

## Collaborative learning approach

The collaborative learning (CL) approach is an educational methodology emphasizing group-based learning, where students work together in small teams or as a whole class to achieve common learning goals (*Supena, Darmuki & Hariyadi, 2021*). It encourages active participation, discussion, and cooperation among learners, fostering a collaborative and supportive learning environment. *Al-Azawi (2018)* showed that combining CL and VR technology enhances student engagement and motivation. VR's immersive and interactive

nature encourages active participation and increases students' interest in the learning process. *vander Meer et al. (2023)* specifies five categories of abilities and competencies often developed using VR and CL (VRCL).

Furthermore, several elements and design concepts are defined in terms of what these settings should provide for developing these talents. VRCL interest's educational sectors and domains due to a desire to innovate, build communities, facilitate distant cooperation, and improve learners' socialization abilities. This research implies that VR can be useful for promoting and improving collaborative learning. According to the findings of *Fischer et al. (2021)*, the primary findings of the work are that the placement of the learner in a virtual classroom impacts the teacher's attention during teaching in virtually immersive and CL settings and social interactions. It also demonstrates that the closeness of the VR device to the teacher in a classroom boosts the learning effect of learners.

## Experiential learning theory

The significance of experience in the educational process is emphasized extensively in the theory of experiential education. According to the notion, learning happens in a cyclical process that includes tangible experiences, contemplative inquiry, and abstraction views (*Heinrich & Green, 2020*). Experiential learning (EL) theory plays a crucial role in designing and implementing VR laboratories. EL and VR technology have tremendous potential to revolutionize education and research (*Asad et al., 2021*). EL theory emphasizes reflection on experiences, and VR can facilitate this by providing opportunities for students to review and analyze their actions within the virtual environment. Artificial intelligence-supported systems can enable personalized learning by adapting to students' needs. By analyzing vast amounts of data, AI can assess students' strengths, weaknesses, and learning patterns (*Murtaza et al., 2022*). This information is used to tailor learning content and activities to meet each student's unique needs. Thanks to artificial intelligence, students can engage in self-directed learning, focus on areas that need improvement, and skip content they already know (*Al Mamun, Lawrie & Wright, 2022*). This personalized approach encourages a deeper understanding of the content and engages students. *Asad et al. (2022)* intends to investigate the impact of VR as a teaching tool for promoting EL among undergraduate students. This was sequential exploratory research, with the first phase being qualitative and the second being quantitative. The acquired data were analyzed individually and utilized to collect more data. The study demonstrates that VR is useful for EL since it immerses pupils. With exposure to VR, students gain a sense of connection and presence in the offered virtual environment. The study of *Kartikasari & Anggaryani (2022)* demonstrates that three creative thinking sessions took place with teams composed of students and faculty members from a variety of disciplines. Three low-fidelity VR models were created during these workshops, assessed and amended with the use of three different student group discussions. According to these results and findings VR application design elements that offer a comprehensive immersive learning process in institutions of higher learning.

## Constructivist learning theory

The constructivist learning theory is a theory of learning that places an emphasis on the learners' active participation in the construction of their own comprehension and

awareness of the world (*Mattar, 2018*). It proposes that learning builds upon existing knowledge, experiences, and mental models rather than passively receiving information from teachers or instructional materials. Learners actively engage with new information and experiences, integrating them with their existing knowledge and prior experiences in constructivist learning. According to the study *Soliman et al. (2021)*, the constructivist learning theory can significantly contribute to developing and designing VR laboratories. In VR laboratories, learners can engage in interactive simulations, experiments, and problem-solving activities, promoting active exploration and EL. AI-powered learning platforms can create a flexible learning environment that changes content, difficulty levels, and feedback based on student interaction and progress (*Srinivasa, Kurni & Saritha, 2022*). This change is based on the view that learning is a continuous process through the learner's experience. AI technologies can create realistic simulations and models that allow students to explore complex and emerging concepts in dynamic and interactive ways (*Hernandez-de Menendez, Escobar Díaz & Morales-Menendez, 2020*). Through these simulated experiences, students can enhance their understanding through experimentation, observation, and a combination of theory and practice. In VR laboratories, learners can control their learning experiences, selecting topics of interest and setting their own pace because the constructivist learning theory emphasizes learner autonomy and individualized learning paths (*Mystakidis, Christopoulos & Pellas, 2022*). The study of *Bahari (2023)* has shown that educators can create dynamic and interactive learning environments that cater to students' needs, foster a deeper understanding of concepts, and equip them with valuable skills for future academic and professional endeavors.

## DISCUSSION

The taxonomy of VR applications developed for this study not only guided our systematic review but also emerged as a crucial tool for understanding the intersection of VR technology and pedagogical theories in higher education. As discussed in the results, the alignment of our findings with the taxonomy's categories validates its utility as a framework for categorizing and evaluating VR applications. Furthermore, the identification of emerging applications suggests the taxonomy's potential for adaptation and expansion, reflecting the dynamic nature of VR technology. This adaptability makes the taxonomy an invaluable resource for educators and researchers seeking to explore innovative VR applications that enhance learning outcomes, as well as for curriculum designers aiming to integrate VR technologies into educational settings effectively.

### Evidence and findings

Using these learning theories and methodologies in VR is quite successful, with favorable results as described in Fig. 9. According to *Monita & Ikhsan (2020)*, Students are more likely to take an interest in and become immersed in learning activities when they use VR, which is one reason why this technology is superior to other forms of multimedia. According to another study *Zulherman et al. (2021)*, VR has the power to entice a person to a new environment and has the potential to increase educational quality by unlocking the ability to learn more than previously. The authors also state that using VR in education

is extremely suited for making courses memorable, so it has become an essential tool. The study of *Chen, Hung & Yeh (2021)* explores how incorporating VR technology into PBL settings influences students' motivation for, ability to solve problems during, and vocabulary acquisition when studying English as a second language. *Drey et al. (2022)* shows that VR allows for creating low-cost, high-quality laboratories and missions to hazardous locations; CL between two people is another VR-independent method for improving learning outcomes. *Jumbri & Ishak (2022)* demonstrates that using VR in learning will favor students' competence, variety, and adaptation. The use of a VR lab increases curiosity, and many students reported that learning became clearer and more pleasurable as a result. *Kee & Zhang (2022)* highlights the potential significance of VR in boosting EL by providing students with hands-on experience in several educational areas using VR technologies to simulate such learning contexts. According to *Moorhouse, tom Dieck & Jung (2019)*, VR is a good tool for applying the constructivist approach and *Meyer, Omdahl & Makransky (2019)* used this theory in their simulator design to improve students' comprehension of astrophysics. The study of *An et al. (2023)* demonstrates that although unfamiliarity with handling the devices might make students uncomfortable, VR substantially influences students' excitement for studying and developing hands-on practical skills. The author analyzes the impacts of VR technology in various phases of environmental art design (*Du, 2021*) and then conducts a more particular investigation based on the various aspects of screen design and the design of interiors. Based on specific facts, analyze the layout and design planning. The all-encompassing mix of VR technology and creative design infuses new vitality into the production procedures.

VR can offer numerous benefits, but it should not be seen as a complete replacement for traditional teaching methods. Instead, it should be integrated strategically to complement and enhance the learning experience, considering individual students' specific educational goals and needs. By all these findings and pieces of evidence, we are in a position to assert with a high level of certainty that combining educational theories with the application of VR will help bridge the gap in the adaption of technology for educational purposes and will provide benefits to students participating in education. The capabilities of VR tools and their integration into education are likely to improve and expand, potentially leading to even more significant benefits for students in the future. The numerous studies that were reviewed present conclusive proof of the benefits.

## Actionable insights for implementing VR in education

– Align VR content with educational goals: Educators should integrate VR content that complements and enhances the curriculum. It is essential to match VR activities with specific learning objectives to ensure educational value.
– Embrace pedagogical frameworks: Design VR experiences to support learning theories such as constructivism and experiential learning. This approach promotes deeper understanding and engagement among students.
– Ensure accessibility and inclusivity: VR experiences must be accessible to all students, including those with disabilities. Developers should focus on creating inclusive content that caters to diverse learning needs and styles.

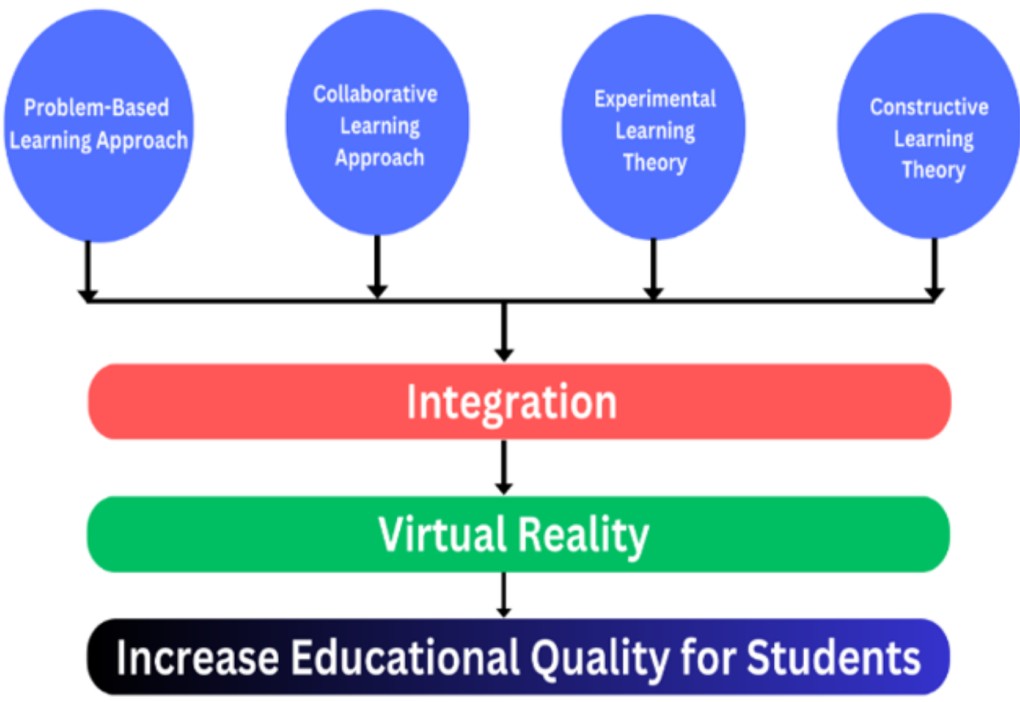

**Figure 9** Integration of educational theories and approaches with VR to get better results.

- Train educators on VR usage: Institutions should offer professional development opportunities for educators. Training on effective VR integration into teaching practices maximizes the technology's benefits.
- Foster collaborative learning opportunities: Use VR to encourage teamwork and collaboration among students. Virtual environments can support group projects and discussions, enhancing the learning experience.
- Utilize VR for personalized learning: Leverage VR's adaptability to offer personalized learning paths. Customized experiences can address individual students' strengths, weaknesses, and interests.
- Promote safe and scalable practical experiences: VR provides a platform for conducting risk-free experiments and simulations. This tool allows students to explore and learn in a safe environment, overcoming traditional lab limitations.
- Encourage active learning: Design VR experiences that require active participation from students. Interactive simulations and problem-solving activities in VR settings engage students more effectively than passive learning methods.
- Implement continuous evaluation: Gather feedback from students and educators on VR learning experiences. Continuous evaluation helps in refining VR content and teaching strategies for better outcomes.
- Stay updated with VR advances: Educators and institutions should keep abreast of the latest developments in VR technology. Staying informed enables the integration of new features and tools that enhance learning experiences.

By incorporating these actionable insights, educators and institutions can effectively harness the potential of VR technology to enrich the educational landscape, making learning more engaging, effective, and inclusive.

## Limitations

It is clear from the extensive research conducted for this article that VR is an effective instructional tool for education. VR is a modern and dynamic medium that necessitates different techniques and methods from traditional or distance learning. VR can provide novel affordances like presence, incarnation, agency, and engagement, but it can also present new obstacles like mental fatigue, sickness from motion, and ethical quandaries. To create, execute, and assess VR educational experiences and outcomes, educators and researchers must develop and use relevant pedagogical ideas, tactics, and technologies. Aside from the more obvious constraints of finance and technical competence, most teachers are unsure of what design and development concerns are critical when preparing to adopt a VR system. VR content, such as modeling, scenarios, and circumstances, is frequently time-consuming and costly to create and maintain. The VR applications, on the other hand, will require thorough planning and design to ensure it is compliant with the course's curriculum, learning objectives, and results. In addition, it is necessary to incorporate learning theories and methods into the design to make the most of the student's educational experience, as well as their cognition and abilities. To ensure immersion, VR content must include excellent visuals, audio, and interaction. Furthermore, VR content must be timely, precise, and diversified to fulfill the requirements and preferences of various learners and circumstances. Educators and content providers must collaborate and harness existing resources and platforms to generate and organize VR content for education.

## Solutions to limitations

Here are some avenues:

– Overcoming technical challenges: Research and development should focus on user-friendly VR hardware and software. Designs need to reduce physical strain. Software must minimize latency to decrease discomfort.

– Pedagogical integration: Professional development programs in VR pedagogy are essential. These programs should cover content creation, curriculum integration, and instructional strategies.

– Cost and resource management: To reduce VR content creation costs, leveraging open-source resources and collaborative platforms is advised. Consortia among educational institutions can share resources and costs, creating shared VR content libraries.

– Design and development guidance: Clear guidelines for VR application design and development are necessary. These guidelines will assist educators in aligning VR experiences with learning objectives and ensuring content relevance.

– Learning theories application: Frameworks that integrate specific learning theories into VR design can enhance learning outcomes. These frameworks should align VR content with theories like constructivism or experiential learning.

- Quality and immersion: Training for content developers in VR production techniques is crucial. Standards for audiovisual fidelity and interactivity must be adopted to ensure immersive quality.
- Collaboration across sectors: Collaboration between educational institutions, VR technology firms, and content creators can lead to innovative content development and platform optimization. This partnership facilitates access to the latest VR technologies and pedagogical insights.

By implementing these solutions, the manuscript aims to address the limitations identified, paving the way for an effective and comprehensive use of VR in education. These initiatives will help overcome barriers, enhancing VR's educational value and ensuring its successful integration into educational practices.

## CONCLUSION AND FUTURE RECOMMENDATIONS

In conclusion, this review highlights VR's transformative role in education when integrated with educational theories and practices. As VR technology evolves, our implementation strategies in educational settings must also progress.

VR creates immersive and interactive learning environments. This makes educational content more engaging and memorable, benefiting both the education sector and students. Students gain access to learning materials in flexible ways, improving their performance. Educational theories offer insights into learning and information processing. Integrating these theories into VR design enables developers to align experiences with learning outcomes. This enhances the effectiveness of educational content. VR labs cater to individual needs and learning styles. They provide personalized content and challenges based on student progress. Students can access these labs remotely, conducting experiments and practicing skills from anywhere. VR labs ensure a safe environment for experiments and simulations. Traditional lab setups are costly, requiring significant investment in equipment and facilities. VR labs reduce these costs substantially.

For professional development and teacher training, VR allows teachers to practice teaching skills, classroom management, and design modeling. Ongoing efforts aim to enhance VR's accessibility and inclusiveness. These efforts address affordability, device compatibility, and access for individuals with learning disabilities. Despite recent advancements, challenges remain in fully leveraging VR in educational environments. Overcoming these challenges is crucial for maximizing VR's potential in education.

Future research directions include:

- Professional development: Explore VR's role in teacher training, focusing on teaching skills and classroom management.
- Accessibility and inclusivity: Address VR's affordability, compatibility, and accessibility for a diverse learner population.
- Integration strategies: Investigate effective VR integration into educational frameworks.
- Long-term effects: Conduct longitudinal studies on VR's impact on learning outcomes and career readiness.

 – Ethical considerations: Examine ethical issues related to VR use in education, including privacy and data security.

Addressing these areas will drive VR advancements in education, supporting both academic and industrial sectors. Early adopters among educational institutions will gain a competitive edge, ensuring educational excellence and quality for their students.

### Funding
The authors received no funding for this work.

### Competing Interests
Habib Hamam is a professor at the Spectrum of Knowledge Production and Skills Development, Sfax, Tunisia.

### Author Contributions
- Fatma Mallek conceived and designed the experiments, performed the experiments, analyzed the data, performed the computation work, prepared figures and/or tables, authored or reviewed drafts of the article, and approved the final draft.
- Tehseen Mazhar conceived and designed the experiments, performed the experiments, analyzed the data, performed the computation work, prepared figures and/or tables, authored or reviewed drafts of the article, and approved the final draft.
- Syed Faisal Abbas Shah conceived and designed the experiments, performed the experiments, analyzed the data, performed the computation work, prepared figures and/or tables, authored or reviewed drafts of the article, and approved the final draft.
- Yazeed Yasin Ghadi conceived and designed the experiments, performed the experiments, analyzed the data, performed the computation work, prepared figures and/or tables, authored or reviewed drafts of the article, and approved the final draft.
- Habib Hamam conceived and designed the experiments, performed the experiments, analyzed the data, performed the computation work, prepared figures and/or tables, authored or reviewed drafts of the article, supervision and funding, and approved the final draft.

### Data Availability
This is a literature review.

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
