# Peer review of "A review on cultivating effective learning: synthesizing educational theories and virtual reality for enhanced educational experiences"

_PeerJ Computer Science, doi:10.7717/peerj-cs.2000_

## Round 0.1 · original submission · Major Revisions

Dear authors,

Thank you for submitting your article. The reviewers’ comments are now available. Your article has not been recommended for publication in its current form. However, we encourage you to address the reviewers' concerns and criticisms and resubmit your article once you have updated it accordingly.

Reviewer 2 has suggested that you cite specific references. You are welcome to add it/them if you believe they are relevant. However, you are not required to include these citations, and if you do not include them, this will not influence my decision.

Best wishes,

**Language Note:** The review process has identified that the English language must be improved. PeerJ can provide language editing services - please contact us at [email protected] for pricing (be sure to provide your manuscript number and title). Alternatively, you should make your own arrangements to improve the language quality and provide details in your response letter. – PeerJ Staff

Reviewer 1 ·

Basic reporting

The review fails to meet standards of a comprehensive review with clear future directions for the readers.

Experimental design

no comments

Validity of the findings

No comments,

Additional comments

In the paper titled “Cultivating Effective Learning: Synthesizing Educational Theories and Virtual Reality for Enhanced Educational Experiences” authors review the impact of immersive technologies for effective learning. This study explores the benefits of utilizing virtual reality (VR) technology in education, particularly in universities and higher education institutions. By integrating VR with educational theories, the aim is to create meaningful and active learning experiences that enhance comprehension. The paper reviews various educational theories emphasizing immersive and interactive learning, assessing their suitability for VR-based applications. The analysis suggests that VR has the potential to revolutionize education by providing immersive experiences that overcome geographical barriers and promote engagement. My concerns and suggestions as below
• The article is a review that is not reflected in the title of the article. It should be updated.
• The abstract continues to discuss the impact of merger of VR with learning without discussion how the review was performed and what are the outcomes.
• The reference at line 43 appears at the wrong place.
• The technical writing is not upto the mark and should be improved to ensure that an international audience can clearly understand and develop interest in the scientific research being presented. Consider sentences at 57-58. Some sentences are starting from word “because”.
• The contributions of the article are not mentioned.
• What is the purpose of section 2 in a review article? The authors should have provided an overview of similar studies rather than going into the history of VR.
• If there is no 4.2m what is the need of 4.1? The article organization needs serious upgrade.
• All figures need to have better pixel quality and uniform color scheme.
• What are the outcomes of the review? There are no meaningful takeaways for the reader.
• The future research directions should be itemized and discussed in more detail.

Reviewer 2 ·

Basic reporting

The article presents a fair comprehensive review of the VR, AR, MR and XR technology for education. 84 reference papers were included in the review.

Experimental design

Yet the study design can be significantly enhanced by including

1) Demographic information regarding the literature review,
2) More comparative information among VR, AR, MR and XR in education, their applications in different higher education areas of science, technology, engineering, arts and mathematics.
3) VR, AR, MR and XR for education in terms of situational interest and self-efficacy.
4) VR Simulation and Serious Game for education

There are quite several important VR/AR studies recently reported (see below for some of them) missed in the review. It will be interesting to show the differences between the proposed review vs. other review articles published.

- The effect of virtual reality on knowledge acquisition and situational interest regarding library orientation in the time of Covid-19, https://doi.org/10.1016/j.acalib.2023.102789
- Self-Efficacy in a 3-Dimensional Virtual Reality Classroom—Initial Teacher Education Students’ Experiences, https://doi.org/10.3390/educsci12060368
- Virtual Reality in Education: A Review of Learning Theories, Approaches and Methodologies for the Last Decade, https://doi.org/10.3390/electronics12132832
- A review of the application of virtual reality technology in higher education based on Web of Science literature data as an example, https://doi.org/10.3389/feduc.2022.1048816
- Mixed Reality for Education, https://link.springer.com/book/10.1007/978-981-99-4958-8
- Virtual and augmented reality, simulation and serious games for education, https://link.springer.com/book/10.1007/978-981-16-1361-6

Validity of the findings

NA

Additional comments

Figure 5 is on teacher-centric education. It will be good to look into problem from student-centric learning which is widely considered a merit of VR in education.

Reviewer 3 ·

Basic reporting

Although the authors have done very well in the manuscript and the work is very good still some points need to be addressed
1- The structure of the paper needs improvement.
1- The introduction section is too small. The author needs to add some data related to the role of AI and ChatGPT in education
2- The author needs to remove Figure 1 and add some other attractive figure
3- The author needs to add a section contribution of the study and organization of the study in the introduction
4- The literature review section is also small and puts some data related to other techniques like ML, DL, and LLM roles in education .


Also, the authors need to pay attention to grammatical mistakes.

Experimental design

5- The methodology section is well organized and presented in good form. Furthermore, figure 2 is the proposed methodology or flow of work.
6- The author needs to present research questions without tables
7- The author needs to add year-wise paper selections that are used for research questions
8- The author needs to add research questions in which they explain the role of VR in different fields

Validity of the findings

The section Evidence and Findings of Discussion needs improvement. Although the authors mentioned some work done applying VR in different fields of education and learning process, they need to show and discuss whether that was successful and if not what they suggest in order to make it successful. Otherwise, the work will be considered as only summarizing what is in the research papers.

Additional comments

Overall paper is very good and my recommendation is minor revision.

·

Basic reporting

The paper structure well managed but there are still some improvements require to fulfil the publication requirements.

1) In Introduction directly started the traditional methods, please write first paragraph to reveal the problem and its importance and then goes towards the other approaches and methods. Second there is need to check the citation place. I think you are maximum details about the topic but need to adjust the paragraph for paper story understanding.

Traditional methods of consuming digital content, such as two-dimensional screens and static interfaces, often limit our ability to engage with the information presented [1] fully.

2) The study is completed or still going? like here future sentence
Specifically, the focus of this overview will be on the use of VR systems in education sector and laboratories

3) please check Grammar issues

Experimental design

There is a need to improve the writing style to understand the study story and mention the importance/purpose of the study. Please write in bullets or with an order to understand the purpose of the study so the reviewer can easily pick the study purpose.

1: Please write a literature review in an academic style; for instance, if there is a discussion about the previous method and the second is also related to that or improved, then relate them with paragraph-started words like similarly, also, further, etc.
2: common mistakes like
In [62], the authors designed
According to the findings of [66],
The study of [69] intends etc
Please check academic writing for it

What is meaning/stand for R.Q here? Reader may be not expert in this domain

The framework's primary objective is maintaining track of the data required to respond to the R.Q.

Validity of the findings

There are still improvements required to meet the criteria, especially in writing style.

Reviewer 5 ·

Basic reporting

The authors have effectively structured the manuscript, primarily concentrating on the pivotal role of VR in the learning process. However, certain issues persist within the text, such as overly lengthy sentences. Personally, I suggest that the authors also incorporate data pertaining to other technologies, such as ChatGPT, LLMs, and the broader influence of AI in education. This would enrich the discussion and provide a more comprehensive understanding of the topic.

Experimental design

The structure of the manuscript suggests that the authors have conducted a Systematic Literature Review (SLR). While the SLR is presented adequately, there is a need for clarification regarding the initial paper selection process, final paper selection criteria, and year-wise selection process. Additionally, the authors should enhance the attractiveness of the PRISMA diagram. The statement regarding the use of four libraries for paper selection and keyword searching requires elaboration for clarity.

Furthermore, it is advisable for the authors to include the research objectives in the introduction section. In addition, the inclusion and exclusion criteria should be outlined in bullet points, clearly indicating the basis on which papers were included or excluded from the study. This will improve transparency and enhance the understanding of the methodology employed.

Validity of the findings

While the authors have effectively addressed the findings pertaining to the research questions, several additional points require attention:

- It is essential for the authors to incorporate some disadvantages associated with both Traditional Learning Methods and VR Learning Methods.
- Furthermore, the manuscript should include the drawbacks of VR Laboratories in Educational Environments, the Problem-Based Learning Approach, and the Collaborative Learning Approach.

Additional comments

- Additionally, it is crucial to discuss the role of AI in the learning process, particularly within frameworks such as the Experiential Learning Theory, Constructivist Learning Theory, and Traditional Learning Methods.
- Lastly, the authors should propose solutions to the limitations mentioned in the manuscript to provide a more comprehensive understanding of the topic and potential avenues for improvement.

---

## Round 0.2 · Minor Revisions

Dear authors,

Thank you for clearly addressing many of the reviewers' comments and suggestions. Although the previous review process has shown that the English language needs to be improved, the paper still needs to be proofread. Please pay particular attention to the reviews of Reviewer 1. Also, some paragraphs are too long to read. They should be split into two or more paragraphs.

Best wishes,

**Language Note:** The Academic Editor has identified that the English language must be improved. PeerJ can provide language editing services - please contact us at [email protected] for pricing (be sure to provide your manuscript number and title). Alternatively, you should make your own arrangements to improve the language quality and provide details in your response letter. – PeerJ Staff

Reviewer 1 ·

Basic reporting

the changes are not up to the mark. several issues highlighted in the initial round of revision have not been carefully considered. For example, I highlighted language issues. In the revised manuscript, three sentences start with because. I can not check the full manuscript for such issue. It is the responsibility of all authors to make sure the language is correct.
Secondly, I mentioned that the article contributions are not listed. In the revised manuscript, the contributions are listed. However, they are contributions of AR/VR in education. They do not reflect the contributions of this article. What does this article contribute to scientific knowledge? any taxonomy? literature review? comparative analysis?
Now the introduction is too long with repetition of ideas. The motivation of studying AR/VR in effective learning is repeated too many times. The introduction should be reduced, concise, to make reader interested in reading the rest of the article
The purpose of section 2 is still not clear. Why their is a section of literature review before research methods?

Experimental design

Nill

Validity of the findings

same as above

Additional comments

same as above

Reviewer 3 ·

Basic reporting

The authors addressed the comments and did changes accordingly (shown in the paper with review tracking)

Experimental design

The study design is now well structured and the authors added several paragraphs to make the contribution clear. The research questions posed by the authors are clearly defined and precisely articulated, providing a solid foundation for the study.

Validity of the findings

All was well explained

Additional comments

The current version of the paper is worth for publishing in the journal

Reviewer 5 ·

Basic reporting

'no comment'

Experimental design

'no comment'

Validity of the findings

'no comment'

Additional comments

'no comment'

---

## Round 0.3 · accepted · Accept

Dear authors,

Thank you for the revision and for clearly addressing all the reviewers' comments. I confirm that the paper is improved and addresses the concerns of the reviewers. Your paper is now acceptable for publication in light of this revision.

Best wishes,

Reviewer 1 ·

Basic reporting

The authors have revised the article carefully now. IT is acceptable in current format.

Experimental design

Nill

Validity of the findings

Same as above